# PositionRank: An Unsupervised Approach to Keyphrase Extraction from Scholarly Documents

## Abstract

The large and growing amounts of online scholarly data present both challenges and opportunities to enhance knowledge discovery. One such challenge is to automatically extract a small set of keyphrases from a document that can accurately describe the document's content and can facilitate fast information processing. In this paper, we propose PositionRank, an unsupervised model for keyphrase extraction from scholarly documents that incorporates information from all positions of a word's occurrences into a biased PageRank. Our model obtains remarkable improvements in performance over PageRank models that do not take into account word positions as well as over strong baselines for this task. Specifically, on several datasets of research papers, PositionRank achieves improvements as high as 29.09%.

## 1 Introduction

The current Scholarly Web contains many millions of scientific documents. For example, Google Scholar is estimated to have more than 100 million documents. On one hand, these rapidly-growing scholarly document collections offer benefits for knowledge discovery, and on the other hand, finding useful information has become very challenging. Keyphrases associated with a document typically provide a high-level topic description of the document and can allow for efficient information processing. In addition, keyphrases are shown to be rich sources of information in many natural language processing and information retrieval tasks such as scientific paper summarization, classification, recommendation, clustering, and search (Abu-Jbara and Radev, 2011; Qazvinian et al., 2010; Jones and Staveley, 1999; Zha, 2002; Zhang et al., 2004; Hammouda et al., 2005). Due to their importance, many approaches to keyphrase extraction have been proposed in the literature along two lines of research: supervised and unsupervised (Hasan and Ng, 2014, 2010).

In the supervised line of research, keyphrase extraction is formulated as a binary classification problem, where candidate phrases are classified as either positive (i.e., keyphrases) or negative (i.e., non-keyphrases) (Frank et al., 1999; Hulth, 2003). Various feature sets and classification algorithms yield different extraction systems. For example, Frank et al. (1999) developed a system that extracts two features for each candidate phrase, i.e., the *tf-idf* of the phrase and its distance from the beginning of the target document, and uses them as input to Naïve Bayes classifiers. Although supervised approaches typically perform better than unsupervised approaches (Kim et al., 2013), the requirement for large human-annotated corpora for each field of study has led to significant attention towards the design of unsupervised approaches.

In the unsupervised line of research, keyphrase extraction is formulated as a ranking problem with graph-based ranking techniques being considered state-of-the-art (Hasan and Ng, 2014). These graph-based techniques construct a word graph from each target document, such that nodes correspond to words and edges correspond to word association patterns. Nodes are then ranked using graph centrality measures such as PageRank (Mihalcea and Tarau, 2004; Liu et al., 2010) or HITS (Litvak and Last, 2008), and the top ranked phrases are returned as keyphrases. Since their introduction, many graph-based extensions have been proposed, which aim at modeling various types of information. For example, Wan and Xiao (2008) proposed a model that incorporates a local neighborhood of the target document correspond-

Factorizing Personalized **Markov Chains** for Next-**Basket Recommendation**
by Steffen Rendle, Christoph Freudenthaler and Lars Schmidt-Thieme

Recommender systems are an important component of many websites. Two of the most popular approaches are based on **matrix factorization** (MF) and **Markov chains** (MC). MF methods learn the general taste of a user by factorizing the matrix over observed user-item preferences. [...] we present a method bringing both approaches together. Our method is based on personalized transition graphs over underlying **Markov chains**. [...] our factorized personalized MC (FPMC) model subsumes both a common **Markov chain** and the normal **matrix factorization** model. [...] we introduce an adaption of the Bayesian Personalized Ranking (BPR) framework for sequential basket data. [...]

Author-input keyphrases: *Basket Recommendation, Markov Chain, Matrix Factorization*

Figure 1: The title and abstract of a WWW paper by Rendle et al. (2010) and the author-input keyphrases for the paper. Red bold phrases represent the gold-standard keyphrases for the document.

ing to its textually-similar documents, computed using the cosine similarity between the *tf-idf* vectors of documents. Liu et al. (2010) assumed a mixture of topics over documents and proposed to use topic models to decompose these topics in order to select keyphrases from all major topics. Keyphrases are then ranked by aggregating the topic-specific scores obtained from several topic-biased PageRanks. We posit that other information can be leveraged that has the potential to improve unsupervised keyphrase extraction.

For example, in a scholarly domain, keyphrases generally occur on positions very close to the beginning of a document and occur frequently. Figure 1 shows an anecdotal example illustrating this behavior using the 2010 best paper award winner in the World Wide Web conference. The author input keyphrases are marked with red bold in the figure. Notice in this example the high frequency of the keyphrase "Markov chain" that occurs very early in the document (even from its title). Hence, *can we design an effective unsupervised approach to keyphrase extraction by jointly exploiting words' position information and their frequency in documents?* We specifically address this question using *research papers* as a case study. The result of this extraction task will aid indexing of documents in digital libraries, and hence, will lead to improved organization, search, retrieval, and recommendation of scientific documents. The importance of keyphrase extraction from research papers is also emphasized by the SemEval Shared Tasks on this topic from 2017[1] and 2010 (Kim et al., 2010). Our contributions are as follows:

- We propose an unsupervised graph-based model, called PositionRank, that incorporates

---
[1] http://alt.qcri.org/semeval2017/task10/

information from all positions of a word's occurrences into a biased PageRank to score keywords that are later used to score and rank keyphrases in research papers.
- We show that PositionRank that aggregates information from all positions of a word's occurrences performs better than a model that uses only the first position of a word.
- We experimentally evaluate PositionRank on three datasets of research papers and show statistically significant improvements over PageRank-based models that do not take into account word positions, as well as over strong baselines for keyphrase extraction.

The rest of the paper is organized as follows. We summarize related work in the next section. PositionRank is described in Section 3. We then present the datasets of research papers, and our experiments and results in Section 4. Finally, we conclude the paper in Section 5.

## 2 Related Work

Many supervised and unsupervised approaches to keyphrase extraction have been proposed in the literature (Hasan and Ng, 2014).

Supervised approaches use annotated documents with "correct" keyphrases to train classifiers for discriminating keyphrases from non-keyphrases for a document. KEA (Frank et al., 1999) and GenEx (Turney, 2000) are two representative supervised approaches with the most important features being the frequency and the position of a phrase in a target document. Hulth (2003) used a combination of lexical and syntactic features such as the collection frequency and the part-of-speech tag of a phrase in conjunction with a bagging technique. Nguyen and Kan

(2007) extended KEA to include features such as the distribution of candidate phrases in different sections of a research paper, and the acronym status of a phrase. In a different work, Medelyan et al. (2009) extended KEA to integrate information from Wikipedia. Lopez and Romary (2010) used bagged decision trees learned from a combination of features including structural features (e.g., the presence of a phrase in particular sections of a document) and lexical features (e.g., the presence of a candidate phrase in WordNet or Wikipedia). Chuang et al. (2012) proposed a model that incorporates a set of statistical and linguistic features (e.g., *tf-idf*, BM25, part-of-speech filters) for identifying descriptive terms in a text. Caragea et al. (2014) designed features based on information available in a document network (such as a citation network) and used them with traditional features in a supervised framework.

In unsupervised approaches, various measures such as *tf-idf* and topic proportions are used to score words, which are later aggregated to obtain scores for phrases (Barker and Cornacchia, 2000; Zhang et al., 2007; Liu et al., 2009). The ranking based on *tf-idf* has been shown to work well in practice (Hasan and Ng, 2014, 2010), despite its simplicity. Graph-based ranking methods and centrality measures are considered state-of-the-art for unsupervised keyphrase extraction. Mihalcea and Tarau (2004) proposed TextRank for scoring keyphrases by applying PageRank on a word graph built from adjacent words within a document. Wan and Xiao (2008) extended TextRank to SingleRank by adding weighted edges between words that co-occur in a window of variable size $w \geq 2$. Textually-similar neighboring documents are included in ExpandRank (Wan and Xiao, 2008) to compute more accurate word co-occurrence information. Gollapalli and Caragea (2014) extended ExpandRank to integrate information from citation networks where papers cite one another.

Several unsupervised approaches leverage word clustering techniques such as grouping candidate words into topics and then, extract one representative keyphrase from each topic (Liu et al., 2009; Bougouin et al., 2013). Liu et al. (2010) extended topic-biased PageRank (Haveliwala, 2003) to kephrase extraction. In particular, they decomposed a document into multiple topics, using topic models, and applied a separate topic-biased PageRank for each topic. The PageRank scores from each topic were then combined into a single score, using as weights the topic proportions returned by topic models for the document.

The best performing keyphrase extraction system in SemEval 2010 (El-Beltagy and Rafea, 2010) used statistical observations such as term frequencies to filter out phrases that are unlikely to be keyphrases. More precisely, thresholding on the frequency of phrases is applied, where the thresholds are estimated from the data. The candidate phrases are then ranked using the *tf-idf* model in conjunction with a boosting factor which aims at reducing the bias towards single word terms. Danesh et al. (2015) computed an initial weight for each phrase based on a combination of statistical heuristics such as the *tf-idf* score and the first position of a phrase in a document. Phrases and their initial weights are then incorporated into a graph-based algorithm which produces the final ranking of keyphrase candidates. Adar and Datta (2015) extracted keyphrases by mining abbreviations from scientific literature and built a semantically hierarchical keyphrase database. Word embedding vectors were also employed to measure the relatedness between words in graph based models (Wang et al., 2014). Many of the above approaches, both supervised and unsupervised, are compared and analyzed in the ACL survey on keyphrase extraction by Hasan and Ng (2014).

In contrast to the above approaches, we propose PositionRank, aimed at capturing both highly frequent words or phrases and their position in a document. Despite that the *relative position* of a word in a document is shown to be a very effective feature in supervised keyphrase extraction (Hulth, 2003; Zhang et al., 2007), to our knowledge, the position information has not been used before in unsupervised methods. The strong contribution of this paper is the design of a position-biased PageRank model that successfully incorporates all positions of a word's occurrences, which is different from supervised models that use only the first position of a word. Our model assigns higher probabilities to words found early on in a document instead of using a uniform distribution over words.

## 3 Proposed Model

In this section, we describe PositionRank, our fully unsupervised, graph-based model, that simultaneously incorporates the position of words and their frequency in a document to compute a biased PageRank score for each candidate word.

Graph-based ranking algorithms such as PageRank (Page et al., 1998) measure the importance of a vertex within a graph by taking into account global information computed recursively from the entire graph. For each word, we compute a weight by aggregating information from all positions of the word's occurrences. This weight is then incorporated into a biased PageRank algorithm in order to assign a different "preference" to each word.

### 3.1 PositionRank

The PositionRank algorithm involves three essential steps: (1) the graph construction at word level; (2) the design of Position-Biased PageRank; and (3) the formation of candidate phrases. These steps are detailed below.

#### 3.1.1 Graph Construction

Let $d$ be a target document for extracting keyphrases. We first apply the part-of-speech filter using the NLP Stanford toolkit and then select as candidate words only nouns and adjectives, similar to previous works (Mihalcea and Tarau, 2004; Wan and Xiao, 2008). We build a word graph $G = (V, E)$ for $d$ such that each unique word that passes the part-of-speech filter corresponds to a node in $G$. Two nodes $v_i$ and $v_j$ are connected by an edge $(v_i, v_j) \in E$ if the words corresponding to these nodes co-occur within a window of $w$ contiguous tokens in the content of $d$. The weight of an edge $(v_i, v_j) \in E$ is computed based on the co-occurrence count of the two words within a window of $w$ successive tokens in $d$. Note that the graph can be constructed both directed and undirected. However, Mihalcea and Tarau (2004) showed that the type of graph used to represent the text does not significantly influence the performance of keyphrase extraction. Hence, in this work, we build undirected graphs.

#### 3.1.2 Position-Biased PageRank

Formally, let $G$ be an undirected graph constructed as above and let $M$ be its adjacency matrix. An element $m_{ij} \in M$ is set to the weight of edge $(v_i, v_j)$ if there exist an edge between nodes $v_i$ and $v_j$, and is set to $0$ otherwise. The PageRank score of a node $v_i$ is recursively computed by summing the normalized scores of nodes $v_j$, which are linked to $v_i$ (as explained below).

Let $S$ denote the vector of PageRank scores, for all $v_i \in V$. The initial values of $S$ are set to $\frac{1}{|V|}$. The PageRank score of each node at step $t+1$, can

then be computed recursively using:

$$S(t + 1) = \widetilde{M} \cdot S(t) \qquad (1)$$

where $\widetilde{M}$ is the normalized form of matrix $M$ with $\widetilde{m_{ij}} \in \widetilde{M}$ defined as:

$$\widetilde{m_{ij}} = \begin{cases} m_{ij} / \sum_{j=1}^{|V|} m_{ij} & \text{if } \sum_{j=1}^{|V|} m_{ij} \neq 0 \\ 0 & \text{otherwise} \end{cases}$$

The PageRank computation can be seen as a Markov Chain process in which nodes represent states and the links between them are the transitions. By recursively applying Eq. (1), we obtain the principal eigenvector, which represents the stationary probability distribution of each state, in our case of each node (Manning et al., 2008).

To ensure that the PageRank (or the random walk) does not get stuck into cycles of the graph, a damping factor $\alpha$ is added to allow the "teleport" operation to another node in the graph. Hence, the computation of $S$ becomes:

$$S = \alpha \cdot \widetilde{M} \cdot S + (1 - \alpha) \cdot \widetilde{p} \qquad (2)$$

where $S$ is the principal eigenvector and $\widetilde{p}$ is a vector of length $|V|$ with all elements $\frac{1}{|V|}$. The vector $\widetilde{p}$ indicates that, being in a node $v_i$, the random walk can jump to any other node in the graph with equal probability. By biasing $\widetilde{p}$, the random walk would prefer nodes that have higher probability in the graph (Haveliwala, 2003).

The idea of PositionRank is to assign larger weights (or probabilities) to words that are found early in a document and are frequent. Specifically, we want to assign a higher probability to a word found on the $2^{nd}$ position as compared to a word found on the $50^{th}$ position in the same document. We weigh each candidate word with its inverse position in the document before any filters are applied. If the same word appears multiple times in the target document, then we sum all its position weights. For example, if a word is found on the following positions: $2^{nd}$, $5^{th}$ and $10^{th}$, its weight is: $\frac{1}{2} + \frac{1}{5} + \frac{1}{10} = \frac{4}{5} = 0.8$. Summing up the position weights for a given word aims to grant more confidence to frequently occurring words by taking into account the position weight of each occurrence. Then, the vector $\widetilde{p}$ is set to the normalized weights for each candidate word as follows:

$$\widetilde{p} = \left[ \frac{p_1}{p_1+p_2+...+p_{|V|}}, \frac{p_2}{p_1+p_2+...+p_{|V|}}, ..., \frac{p_{|V|}}{p_1+p_2+...+p_{|V|}} \right]$$

The PageRank score of a vertex $v_i$, i.e., $S(v_i)$, can be obtained in an algebraic way by recursively computing the following equation:

$$S(v_i) = (1 - \alpha) \cdot \widetilde{p}_i + \alpha \cdot \sum_{v_j \in Adj(v_i)} \frac{w_{ji}}{O(v_j)} S(v_j)$$

where $O(v_j) = \sum_{v_k \in Adj(v_j)} w_{jk}$ and $\widetilde{p}_i$ is the weight found in the vector $\widetilde{p}$ for vertex $v_i$.

In our experiments, the words' PageRank scores are recursively computed until the difference between two consecutive iterations is less than 0.001 or a number of 100 iterations is reached.

### 3.1.3 Forming Candidate Phrases

Candidate words that have contiguous positions in a document are concatenated into phrases. We consider noun phrases that match the regular expression (adjective)*(noun)+, of length up to three, (i.e., unigrams, bigrams, and trigrams).

Finally, phrases are scored by using the sum of scores of individual words that comprise the phrase (Wan and Xiao, 2008). The top-scoring phrases are output as predictions (i.e., the predicted keyphrases for the document).

## 4 Experiments and Results

### 4.1 Datasets and Evaluation Metrics

In order to evaluate the performance of PositionRank, we carried out experiments on three datasets. The first and second datasets were made available by Gollapalli and Caragea (2014). These datasets were compiled from the CiteSeerX digital library (Giles et al., 1998) and consist of research papers from the ACM Conference on Knowledge Discovery and Data Mining (KDD) and the World Wide Web Conference (WWW). The third dataset was made available by Nguyen and Kan (2007) and consist of research papers from various disciplines. In experiments, we use the title and abstract of each paper to extract keyphrases. The author-input keyphrases are used as gold-standard for evaluation. All three datasets are summarized in Table 1, which shows the number of papers in each dataset, the total number of keyphrases (Kp), the average number of keyphrases per document (AvgKp), and a brief insight into the length and number of available keyphrases.

**Evaluation Metrics.** We use mean reciprocal rank (MRR) curves to illustrate our experimental findings. MRR gives the averaged ranking of the first correct prediction and is defined as:

$$MRR = \frac{1}{|D|} \sum_{d \in D} \frac{1}{r_d}$$

where $D$ is the collection of documents and $r_d$ is the rank at which the first correct keyphrase of document $d$ was found. We also summarize the results in terms of Precision, Recall, and F1-score in a table to contrast PositionRank with previous models since these metrics are widely used in previous works (Hulth, 2003; Wan and Xiao, 2008; Mihalcea and Tarau, 2004; Hasan and Ng, 2014). To compute "performance@$k$" (such as MRR@$k$), we examine the top-$k$ predictions (with $k$ ranging from 1 to 10). We use *average $k$* to refer to the average number of keyphrases for a particular dataset as listed in Table 1. For example, *average $k = 5$* for the WWW dataset. For comparison purposes, we used Porter Stemmer to reduce both predicted and gold keyphrases to a base form.

### 4.2 Results and Discussion

Our experiments are organized around several questions, which are discussed below.

*How sensitive is PositionRank to its parameters?* One parameter of our model that can influence its performance is the window size $w$, which determines how edges are added between candidate words in the graph. We experimented with values of $w$ ranging from 2 to 10 in steps of 1 and chose several configurations for illustration. Figure 2 shows the MRR curves of PositionRank for different values of $w$, on all three datasets. As can be seen from the figure, the performance of our model does not change significantly as $w$ changes.

In addition to the window size, our model has one more parameter, i.e., the damping factor $\alpha$. In order to understand its influence on the performance of PositionRank, we experimented with several values of $\alpha$, e.g., $0.75, 0.8, 0.85, 0.9$, and did not find significant differences in the performance of PositionRank (results not shown due to highly overlapping curves). Hence, in Equation 2, we set $\alpha = 0.85$ as in (Haveliwala, 2003).

*What is the impact of aggregating information from all positions of a word over using a word's first position only?* In this experiment, we analyze the influence that position-weighted frequent words in a document would have on the performance of PositionRank. Specifically, we compare the performance of the model that aggregates information from all positions of a word's occurrences, referred as PositionRank - *full model* with that of the model that uses only the first position

| Dataset | #Docs | Kp | AvgKp | unigrams | bigrams | trigrams | n-grams ($n \geq 4$) |
|---------|-------|------|-------|----------|---------|----------|-----------------------|
| KDD | 834 | 3093 | 3.70 | 810 | 1770 | 471 | 42 |
| WWW | 1350 | 6405 | 4.74 | 2254 | 3139 | 931 | 81 |
| Nguyen | 211 | 882 | 4.18 | 260 | 457 | 132 | 33 |

Table 1: A summary of our datasets.

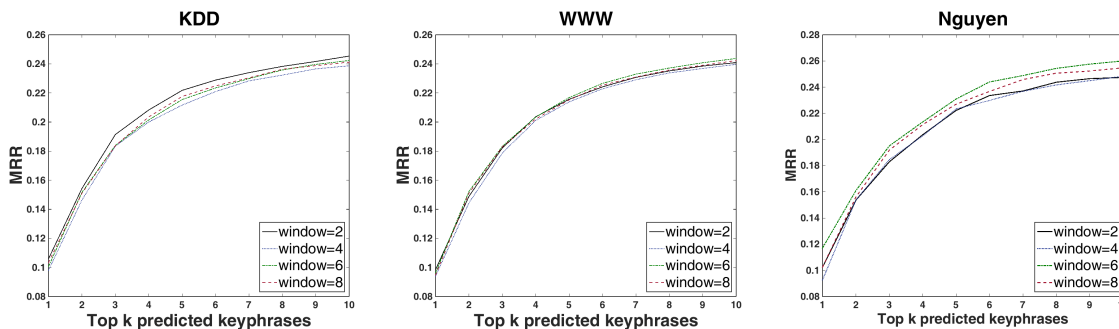

Figure 2: MRR curves for PositionRank that uses different values for the window size.

of a word, referred as PositionRank - *fp*. In the example from the previous section, a word occurring on positions $2^{nd}$, $5^{th}$, and $10^{th}$ will have a weight of $\frac{1}{2} + \frac{1}{5} + \frac{1}{10} = \frac{4}{5} = 0.8$ in the *full model*, and a weight of $\frac{1}{2} = 0.5$ in the *first position (fp)* model. Note that the weights of words are normalized before they are used in the biased PageRank.

Figure 3 shows the results of this experiment in terms of MRR for the top $k$ predicted keyphrases, with $k$ from 1 to 10, for all datasets, KDD, WWW, and Nguyen. As we can see from the figure, the performance of PositionRank - *full model* consistently outperforms its counterpart that uses the *first position* only, on all datasets. We can conclude from this experiment that aggregating information from all occurrences of a word acts as an important component in PositionRank. Hence, we use PositionRank - *full model* for further comparisons.

***How well does position information aid in unsupervised keyphrase extraction from research papers?*** In this experiment, we compare our position-biased PageRank model (PositionRank) with two PageRank-based models, TextRank and SingleRank, that do not make use of the position information. In TextRank, an undirected graph is built for each target paper, so that nodes correspond to words and edges are drawn between two words that occur next to each other in text, i.e., the window size $w$ is 2. SingleRank extends TextRank by adding edges between two words that co-occur in a window of $w \geq 2$ contiguous words in text.

Figure 4 shows the MRR curves comparing PositionRank with TextRank and SingleRank. As

can be seen from the figure, PositionRank substantially outperforms both TextRank and SingleRank on all three datasets, illustrating that the words' positions contain significant hints that aid the keyphrase extraction task. PositionRank can successfully harness this information in an unsupervised setting to obtain good improvements in the extraction performance. For example, PositionRank that uses information from all positions of a word's occurrences yields improvements in MRR@average $k$ of 17.46% for KDD, 20.18% for WWW, and 17.03% for Nguyen over SingleRank.

***How does PositionRank compare with other existing state-of-the-art methods?*** In Figure 5, we compare PositionRank with several strong baselines: TF-IDF, ExpandRank, and TopicalPageRank (TPR) (Hasan and Ng, 2014; Wan and Xiao, 2008; Liu et al., 2010). We selected these baselines based on the ACL survey on keyphrase extraction by Hasan and Ng (2014). In TF-IDF, we calculate the *tf* score of each candidate word in the target document, whereas the *idf* component is estimated from all three datasets. In ExpandRank, we build an undirected graph from each paper and its local textual neighborhood and calculate the candidate words' importance scores using PageRank. We performed experiments with various numbers of textually-similar neighbors and present the best results for each dataset. In TPR, we build an undirected graph using information from the target paper. We then perform topic decomposition of the target document using topic models to infer the topic distribution

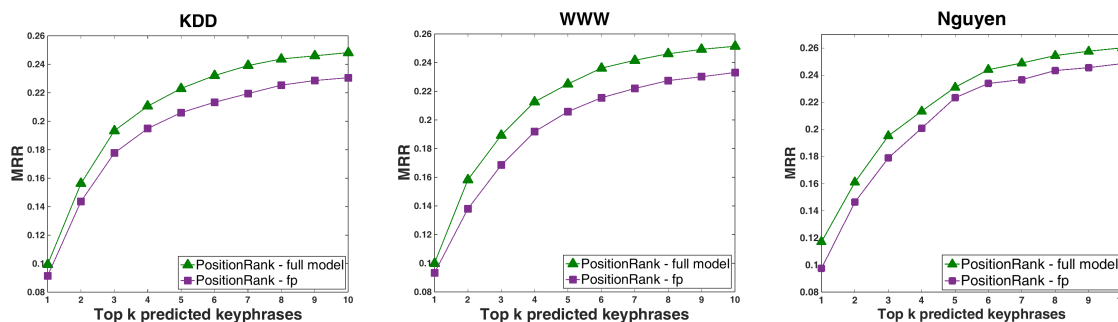

Figure 3: The comparison of PositionRank that aggregates information from all positions of a word's occurrences (full model) with the PositionRank that uses only the first position of a word (fp).

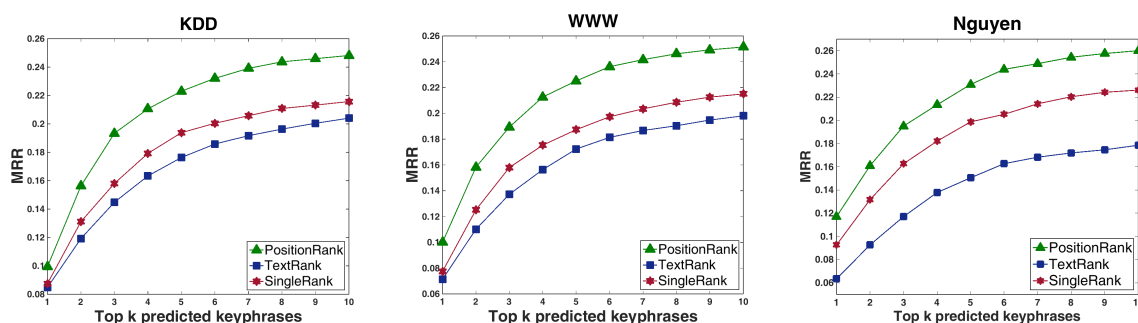

Figure 4: MRR curves for PositionRank and two unbiased PageRank-based models that do not consider position information.

of a document and to compute the probability of words in these topics. Last, we calculate the candidate words' importance scores by aggregating the scores from several topic-biased PageRanks (one PageRank per topic). We used the implementation of topic models from Mallet.[2] To train the topic model, we used a subset of about $45,000$ paper abstracts extracted from CiteSeerX. For all models, the score of a phrase is obtained by summing the score of the constituent words in the phrase.

From Figure 5, we can see that PositionRank achieves a significant increase in MRR over the baselines, on all datasets. For example, the highest relative improvement in MRR@average $k$ for this experiment is as high as $29.09\%$ achieved on the Nguyen collection. Among all models compared in Figure 5, ExpandRank is clearly the best performing baseline, while TPR achieves the lowest MRR values, on all datasets.

### 4.3 Overall Performance

As already mentioned, prior works on keyphrase extraction report results also in terms of precision (P), recall (R), and F1-score (F1) (Hulth, 2003; Hasan and Ng, 2010; Liu et al., 2010; Wan and

Xiao, 2008). Consistent with these works, in Table 2, we show the results of the comparison of PositionRank with all baselines, in terms of P, R and F1 for top $k = 2, 4, 6, 8$ predicted keyphrases, on all three datasets. As can be seen from the table, PositionRank outperforms all baselines, on all datasets. For example, on WWW at top 6 predicted keyphrases, PositionRank achieves an F1-score of $12.1\%$ as compared to $11.2\%$ achieved by ExpandRank and $10.7\%$ achieved by both TF-IDF and TPR. From the table, we can also see that ExpandRank is generally the best performing baseline on all datasets. However, it is interesting to note that, unlike PositionRank that uses information only from the target paper, ExpandRank adds external information from a textually-similar neighborhood of the target paper, and hence, is computationally more expensive.

PositionRank-*first position only* (*fp*) typically performs worse than PositionRank-*full model*, but it still outperforms the baseline methods for most top $k$ predicted keyphrases, on all datasets. For example, on Nguyen at top 4, PositionRank-*fp* achieves an F1-score of $10.5\%$ compared to the best baseline (TF-IDF in this case), which reaches only a score of $9.6\%$.

---

[2]http://mallet.cs.umass.edu/

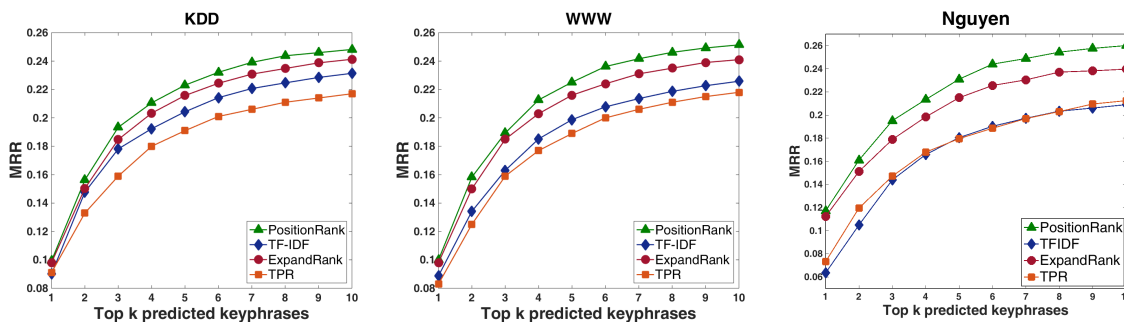

Figure 5: MRR curves for PositionRank and baselines on the three datasets.

| Dataset | Unsupervised method | Top2 | | | Top4 | | | Top6 | | | Top8 | | |
|---|---|---|---|---|---|---|---|---|---|---|---|---|---|
| | | P% | R% | F1% | P% | R% | F1% | P% | R% | F1% | P% | R% | F1% |
| KDD | PositionRank | **11.1** | **5.6** | **7.3** | **10.8** | **11.1** | **10.6** | **9.8** | **15.3** | **11.6** | **9.2** | **18.9** | **12.1** |
| | PositionRank-fp | 10.3 | 5.3 | 6.8 | 10.2 | 10.4 | 10.0 | 9.1 | 13.8 | 10.9 | 8.6 | 17.2 | 11.3 |
| | TF-IDF | 10.5 | 5.2 | 6.8 | 9.6 | 9.7 | 9.4 | 9.2 | 13.8 | 10.7 | 8.7 | 17.4 | 11.3 |
| | TextRank | 8.1 | 4.0 | 5.3 | 8.3 | 8.5 | 8.1 | 8.1 | 12.3 | 9.4 | 7.6 | 15.3 | 9.8 |
| | SingleRank | 9.1 | 4.6 | 6.0 | 9.3 | 9.4 | 9.0 | 8.7 | 13.1 | 10.1 | 8.1 | 16.4 | 10.6 |
| | ExpandRank | 10.3 | 5.5 | 6.9 | 10.4 | 10.7 | 10.1 | 9.2 | 14.5 | 10.9 | 8.4 | 17.5 | 11.0 |
| | TPR | 9.3 | 4.8 | 6.2 | 9.1 | 9.3 | 8.9 | 8.8 | 13.4 | 10.3 | 8.0 | 16.2 | 10.4 |
| WWW | PositionRank | **11.3** | **5.3** | **7.0** | **11.3** | **10.5** | **10.5** | **10.8** | **14.9** | **12.1** | **9.9** | **18.1** | **12.3** |
| | PositionRank-fp | 9.6 | 4.5 | 6.0 | 10.3 | 9.6 | 9.6 | 10.1 | 13.8 | 11.2 | 9.4 | 17.2 | 11.7 |
| | TF-IDF | 9.5 | 4.5 | 5.9 | 10.0 | 9.3 | 9.3 | 9.6 | 13.3 | 10.7 | 9.1 | 16.8 | 11.4 |
| | TextRank | 7.7 | 3.7 | 4.8 | 8.6 | 7.9 | 8.0 | 8.1 | 12.3 | 9.8 | 8.2 | 15.2 | 10.2 |
| | SingleRank | 9.1 | 4.2 | 5.6 | 9.6 | 8.9 | 8.9 | 9.3 | 13.0 | 10.5 | 8.8 | 16.3 | 11.0 |
| | ExpandRank | 10.4 | 5.3 | 6.7 | 10.4 | 10.6 | 10.1 | 9.5 | 14.7 | 11.2 | 8.6 | 17.7 | 11.2 |
| | TPR | 8.8 | 4.2 | 5.5 | 9.6 | 8.9 | 8.9 | 9.5 | 13.2 | 10.7 | 9.0 | 16.5 | 11.2 |
| Nguyen | PositionRank | **10.5** | **5.8** | **7.3** | **10.6** | **11.4** | **10.7** | **11.0** | **17.2** | **13.0** | **10.2** | **21.1** | **13.5** |
| | PositionRank-fp | 10.0 | 5.4 | 6.8 | 10.4 | 11.1 | 10.5 | 11.2 | 17.4 | 13.2 | 10.1 | 21.2 | 13.3 |
| | TF-IDF | 7.3 | 4.0 | 5.0 | 9.5 | 10.3 | 9.6 | 9.1 | 14.4 | 10.9 | 8.9 | 18.9 | 11.8 |
| | TextRank | 6.3 | 3.6 | 4.5 | 7.4 | 7.4 | 7.2 | 7.8 | 11.9 | 9.1 | 7.2 | 14.8 | 9.4 |
| | SingleRank | 9.0 | 5.2 | 6.4 | 9.5 | 9.9 | 9.4 | 9.2 | 14.5 | 11.0 | 8.9 | 18.3 | 11.6 |
| | ExpandRank | 9.5 | 5.3 | 6.6 | 9.5 | 10.2 | 9.5 | 9.1 | 14.4 | 10.8 | 8.7 | 18.3 | 11.4 |
| | TPR | 8.7 | 4.9 | 6.1 | 9.1 | 9.5 | 9.0 | 8.8 | 13.8 | 10.5 | 8.8 | 18.0 | 11.5 |

Table 2: PositionRank against baselines in terms of Precision, Recall and F1-score. Best results are shown in **bold blue**.

A striking observation is that PositionRank outperforms TPR on all datasets. Compared with our model, TPR is a very complex model, which uses topic models to learn topics of words and infer the topic proportion of documents. Additionally, TPR has more parameters (e.g., the number of topics) that need to be tuned separately for each dataset. PositionRank is much less complex, it does not require an additional dataset (e.g., to train a topic model) and its performance is better than that of TPR. TF-IDF and ExpandRank are the best performing baselines, on all datasets. For example, on KDD at $k = 4$, TF-IDF and ExpandRank yield an F1-score of 9.4% and 10.1%, respectively, compared with 8.4%, 9.0% and 8.9% achieved by TextRank, SingleRank and TPR, respectively.

With a paired t-test on our results, we found that the improvements in MRR, precision, recall, and F1-score for PositionRank are statistically significant ($p$-values $< 0.05$).

## 5 Conclusion and Future Work

We proposed a novel unsupervised graph-based algorithm, called PositionRank, which incorporates both the position of words and their frequency in a document into a biased PageRank. To our knowledge, we are the first to integrate the position information in novel ways in unsupervised keyphrase extraction. Specifically, unlike supervised approaches that use only the first position information, we showed that modeling the entire distribution of positions for a word outperforms models that use only the first position. Our experiments on three datasets of research papers show that our proposed model achieves better results than strong baselines, with relative improvements in performance as high as $29.09\%$.

In the future, it would be interesting to explore the performance of PositionRank on other types of documents, e.g., web pages and emails.

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
