# Peer review of "PositionRank: An Unsupervised Approach to Keyphrase Extraction from Scholarly Documents"

_ACL 2017 — decision unknown_

[Official Review · Reviewer 1 · rating 4 · confidence 4]
soundness 5 · originality 3 · clarity 5 · impact 3 · substance 4 · appropriateness 5 · meaningful comparison 4 · presentation format Poster

- Strengths:
Nicely written and understandable.
Clearly organized. Targeted answering of research questions, based on 
different experiments.

- Weaknesses:
Minimal novelty. The "first sentence" heuristic has been in the summarization
literature for many years. This work essentially applies this heuristic
(evolved) in the keyword extraction setting. This is NOT to say that the work
is trivial: it is just not really novel.

Lack of state-of-the-art/very recent methods. The experiment on the system
evaluation vs state-of-the-art systems simply uses strong baselines. Even
though the experiment answers the question "does it perform better than
baselines?", I am not confident it illustrates that the system performs better
than the current state-of-the-art. This somewhat reduces the value of the
paper.

- General Discussion:
Overall the paper is good and I propose that it be published and presented. 

On the other hand, I would propose that the authors position themselves (and
the system performance) with respect to:
Martinez‐Romo, Juan, Lourdes Araujo, and Andres Duque Fernandez. "SemGraph:
Extracting keyphrases following a novel semantic graph‐based approach."
Journal of the Association for Information Science and Technology 67.1 (2016):
71-82.
(with which the work holds remarkable resemblance in some points)

Le, Tho Thi Ngoc, Minh Le Nguyen, and Akira Shimazu. "Unsupervised Keyphrase
Extraction: Introducing New Kinds of Words to Keyphrases." Australasian Joint
Conference on Artificial Intelligence. Springer International Publishing, 2016.